# AnaCP: Toward Upper-Bound Continual Learning via Analytic Contrastive Projection

**Saleh Momeni**[1]**, Changnan Xiao**[2]**, Bing Liu**[1]
[1] Department of Computer Science, University of Illinois Chicago
[2] ChangnXX.github.io
{smomen3, liub}@uic.edu    changnanxiao@gmail.com

## Abstract

This paper studies the problem of class-incremental learning (CIL), a core setting within continual learning where a model learns a sequence of tasks, each containing a distinct set of classes. Traditional CIL methods, which do not leverage pre-trained models (PTMs), suffer from catastrophic forgetting (CF) due to the need to incrementally learn both feature representations and the classifier. The integration of PTMs into CIL has recently led to efficient approaches that treat the PTM as a fixed feature extractor combined with analytic classifiers, achieving state-of-the-art performance. However, they still face a major limitation: the inability to continually adapt feature representations to best suit the CIL tasks, leading to suboptimal performance. To address this, we propose AnaCP (Analytic Contrastive Projection), a novel method that preserves the efficiency of analytic classifiers while enabling incremental feature adaptation without gradient-based training, thereby eliminating the CF caused by gradient updates. Our experiments show that AnaCP not only outperforms existing baselines but also achieves the accuracy level of joint training, which is regarded as the upper bound of CIL.

## 1 Introduction

Continual learning (CL) learns a sequence of tasks incrementally, where each task introduces a set of new classes [1]. The key challenge in CL is catastrophic forgetting (CF) [2], a phenomenon where learning new tasks degrades the performance on previously learned ones. Among CL settings, class-incremental learning (CIL) [3] is particularly challenging as it needs to build a unified classifier to recognize all encountered classes, without task-id information at test time. This paper focuses on the CIL setting. Early CIL approaches typically trained models from scratch using regularization, experience replay, or architectural modifications to mitigate CF [4, 5]. However, they fall significantly short of **joint training** accuracy, which trains a model with the data from all tasks simultaneously, representing the **upper bound** for CIL performance [6].

Recent CIL methods increasingly exploit pre-trained models (PTMs) to improve accuracy by leveraging their strong feature representations [7, 8, 9]. Existing PTM-based approaches for CIL can be broadly divided into three main categories: (i) fine-tuning the PTM, either fully or through lightweight adapter [10], (ii) optimizing learnable prompts while keeping the PTM parameters frozen [11], and (iii) using the PTM as a fixed feature extractor paired with an analytic classifier, such as the nearest class mean (NCM) method or more advanced variants [12, 13, 14]. We use the term **analytic** to describe methods with a **closed-form solution** that requires no gradient-based training. The first two groups require task-specific training, which is slow and prone to CF. The third group is highly efficient and immune to CF because it avoids gradient-based updates.

Analytic approaches often achieve state-of-the-art performance in CIL, outperforming methods that fine-tune the PTM or learn prompts [15, 14]. However, a **major limitation** of the analytic methods

39th Conference on Neural Information Processing Systems (NeurIPS 2025).

is their inability to adapt the PTM features to suit the downstream tasks, leading to suboptimal performance. Prior works have proposed to fine-tune the PTM on the first task, called first session adaptation (FSA) [16, 13]. However, the PTM must remain frozen afterward to maintain the integrity of analytic learning; otherwise, their closed-form solutions will not work.

This paper introduces **AnaCP** (Analytic Contrastive Projection), a novel method that enables analytic approaches to also perform feature adaptation. Our analytic method is related to extreme learning machines (ELMs) [17, 18], as outlined in Section 3. AnaCP adapts features using a contrastive projection layer, inspired by contrastive learning [19, 20], which draws samples from the same class closer while pushing different classes apart to enhance separation. However, unlike standard contrastive learning, which requires gradient-based training for each task and suffers from CF, AnaCP achieves a similar effect analytically. With the adapted features, an NCM classifier can be easily constructed (see Section 4.3). However, we found that building another ELM classifier by sampling feature representations from distributions of previous tasks yields better performance, while remaining fully analytic. To summarize, this work makes the following contributions:

1. We propose a novel analytic approach that enables continual adaptation of feature representations across tasks, addressing the long-standing limitation of fixed representations in prior analytic methods.

2. Our method avoids CF entirely, as it requires no gradient-based training and instead relies on closed-form updates.

3. This approach is highly efficient, requiring no trainable parameters and incurring negligible computational overhead.

4. Through extensive experiments, we show that when paired with a strong PTM, our method achieves accuracy comparable to the joint training upper bound. This is particularly notable, as the inability to match joint training remains a key barrier to the practical adoption of CL.

## 2    Related work

Many CIL approaches have been proposed to deal with CF. These methods can be categorized into three main groups: (i) *Regularization* methods, which mitigate CF by penalizing updates to parameters that are important for old tasks [21, 22, 23, 24], or by aligning the current model with those of earlier ones using knowledge distillation [25, 26]. (ii) *Experience replay* methods, which retain a subset of past task samples in a buffer and use them in the new task training [27, 28, 29, 30, 31].A variation of this is *pseudo-replay*, where synthetic samples or feature representations are generated to simulate past data [32, 33, 34, 35]. Our method also replays feature representations in its final step, but they are sampled from each class's distribution represented by its mean and a shared covariance. (iii) *architecture-based* methods, which expand the network for each task [36, 37, 38], or isolate parameters through learning task-specific sub-networks via masking or orthogonal projections [39, 40, 41, 42, 43], and then use a task-id prediction method at test time [6, 44, 45, 46].

Early approaches to CIL learned both feature representations and classification parameters for each new task without using PTMs, suffering from serious CF. Recent methods address this limitation by leveraging PTMs, which substantially improve CIL performance [7, 12, 47, 13, 6, 14], primarily following three strategies: (i) fine-tuning the PTM, either by directly updating its parameters (e.g., SLCA [10]) or by adding adapters [6, 48, 49]. (ii) learning prompts, where the PTM remains frozen and trainable prompts are introduced to condition its representations (e.g., L2P [11], DualPrompt [47], CODA-Prompt [50], as well as other variants [51, 52, 53]). (iii) treating the PTM as a frozen feature extractor with an analytic classifier [13, 54, 14].

Our method aligns with the third strategy, which relies on closed-form solutions. The simplest method is NCM [55, 56], which computes a mean vector for each class and assigns each test sample to the class with the nearest mean. Another method is SLDA [57], which takes advantage of linear discriminant analysis (LDA) [58]. KLDA [14] applies an RBF kernel to expand the feature space before using LDA for classification. The regularized least squares used in ACIL [12] and GACL [54] is also within this paradigm. Analytic CIL methods can be enhanced by adopting the FSA strategy, where the PTM is fine-tuned on the first task and then kept frozen for subsequent tasks. APER [56] adopts this strategy with an NCM classifier. RanPac [13] has a similar approach but applies a random projection [18] to the feature space, improving class separation. LoRanPAC [59] enables RanPAC

to use a much higher dimensional random projection. FeCAM [15] uses a Mahalanobis distance classifier with normalized covariance matrices.

We follow the analytic CIL paradigm by using PTMs as frozen feature extractors. However, unlike previous methods, we introduce a contrastive projection layer to adapt the features for each task, which enables AnaCP to improve the accuracy while preserving the computational efficiency.

## 3 Background

**Class-Incremental Learning:** CIL seeks to sequentially learn a series of tasks, each introducing a new set of classes, while prohibiting access to data from previously encountered tasks. Specifically, each task $t$ is defined by a training dataset $\mathcal{D}_t = \{(x_t^{(i)}, y_t^{(i)})\}_{i=1}^{n_t}$, where $x_t^{(i)} \in \mathcal{X}_t$ represents an input sample, and $y_t^{(i)} \in \mathcal{Y}_t$ is its class label. The class sets $\mathcal{Y}_t$ are mutually exclusive, and $\mathcal{Y} = \bigcup_{t=1}^{T} \mathcal{Y}_t$ defines the complete set of classes encountered so far. The core objective of CIL is to learn a single classifier $f : \mathcal{X} \to \mathcal{Y}$ capable of predicting the class label of any test instance without the knowledge of the task it belongs to.

**Ridge Regression:** Ridge regression is a closed-form method for training a linear classifier on fixed features using one-hot encoded targets [60]. Given a feature matrix $\mathbf{X} \in \mathbb{R}^{N \times d}$ from the PTM and a one-hot label matrix $\mathbf{Y} \in \mathbb{R}^{N \times C}$, the objective is to minimize the squared error with $L_2$ regularization:

$$\min_{\mathbf{W}} \ \|\mathbf{XW} - \mathbf{Y}\|^2 + \lambda\|\mathbf{W}\|^2, \tag{1}$$

where $\lambda$ controls the strength of the regularization. This objective penalizes deviations from the target labels while encouraging smaller weights to prevent overfitting. The solution is obtained by setting the gradient for $\mathbf{W}$ to zero, yielding the closed-form optimal weights:

$$\mathbf{W} = (\mathbf{X}^\top\mathbf{X} + \lambda\mathbf{I})^{-1}\mathbf{X}^\top\mathbf{Y}, \tag{2}$$

often called a ridge classifier or regularized least squares. Here, $\mathbf{X}^\top\mathbf{X}$ is called the Gram matrix ($\mathbf{G}$) and $\mathbf{X}^\top\mathbf{Y}$ is the cross matrix ($\mathbf{H}$). This closed-form solution is attractive because it forgoes iterative training and directly computes a global optimum.

**Analytic CIL:** When tasks arrive sequentially, we can compute the ridge regression solution without revisiting past data [17, 12]. Let $(\mathbf{X}_t, \mathbf{Y}_t)$ denote the data from task $t$. Rather than storing all data, we can incrementally update the Gram and cross matrices as follows:

$$\mathbf{G}_t = \mathbf{G}_{t-1} + \mathbf{X}_t^\top\mathbf{X}_t, \quad \mathbf{H}_t = \mathbf{H}_{t-1} + \mathbf{X}_t^\top\mathbf{Y}_t, \tag{3}$$

where $\mathbf{G}_t$ and $\mathbf{H}_t$ are the gram and cross matrices after task $t$, respectively. Here, we slightly abuse the notation for $\mathbf{H}_t$ by allowing the number of classes to increase over time. To maintain dimensional consistency, both $\mathbf{H}_{t-1}$ and the one-hot labels in $\mathbf{Y}_t$ must be zero-padded. This allows us to compute the updated solution:

$$\mathbf{W}_t = (\mathbf{G}_t + \lambda\mathbf{I})^{-1}\mathbf{H}_t \tag{4}$$

**Random Projection and Extreme Learning Machines:** To improve the feature representations, the original $d$-dimensional features can be projected into a higher-dimensional space of dimension $D$ before learning the analytic classifier in the projected space. [18] proposes to project the features via a random nonlinear layer:

$$\mathbf{Z} = \phi(\mathbf{XR}), \tag{5}$$

where $\mathbf{R} \in \mathbb{R}^{d \times D}$ is a fixed matrix with random values and $\phi(\cdot)$ is a nonlinear activation, for which we use a GELU function. The classifier is learned analytically via ridge regression:

$$\mathbf{W} = (\mathbf{Z}^\top\mathbf{Z} + \lambda\mathbf{I})^{-1}\mathbf{Z}^\top\mathbf{Y} \tag{6}$$

This solution can be updated incrementally as before (Eqs. 3 and 4). The random projection acts like a kernel function, expanding the feature space and introducing nonlinear interactions among input features, which enhances class separability.

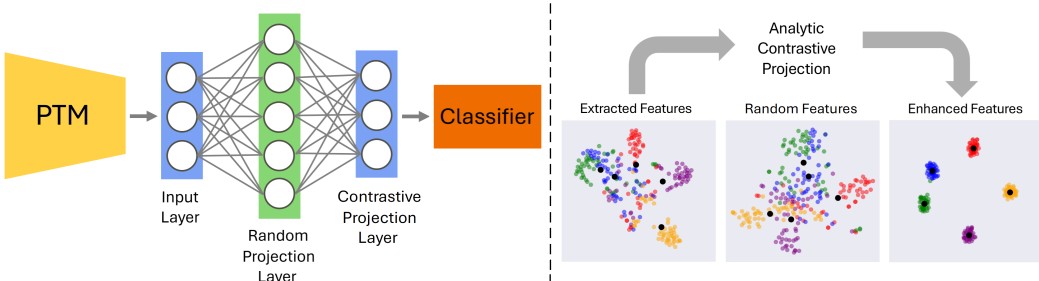

Figure 1: (**Left**) Architecture of AnaCP: The random projection layer uses fixed, randomly assigned weights, while the contrastive projection layer weights are computed analytically. Compared to the ELM architecture, AnaCP introduces an additional contrastive projection layer that adapts the feature representations. (**Right**) $t$-SNE visualization of the input features (first five classes of ImageNet-R) and their enhancement after contrastive projection using DINO-v2 as the PTM. Random features are generally more separable than input features due to their higher dimensionality, even though this is not easily observable in the 2D t-SNE map.

## 4    Methodology

Existing analytic CIL methods operate on fixed feature representations, allowing the classifier to be updated analytically without revisiting prior data. Freezing the PTM is essential because modifying it during training would alter the feature representations, undermining previous analytic updates and leading to CF. However, this rigid strategy also limits the capability to adapt feature representations to suit the downstream CIL tasks.

To overcome this limitation, we propose a novel analytic approach that also adapts feature representations for the continual learning tasks. The pipeline of our approach, illustrated in Figure 1 (left), begins with the PTM generating feature representations, termed the **input layer**. These features are transformed via a **random projection (RP) layer** into a higher-dimensional space [18] and then passed through the **contrastive projection (CP) layer**, which enhances the class separability. The CP layer draws inspiration from contrastive learning, where the samples from the same class (**positives**) are drawn closer, while samples from different classes (**negatives**) are pushed apart, typically using a contrastive loss. However, applying this directly to CIL would lead to CF due to iterative gradient updates. We achieve similar effects via an analytic CP layer that can be incrementally updated as new tasks arrive. The impact of this CP layer on feature distribution is depicted in Figure 1 (right). Finally, a **classifier layer** makes the final predictions based on these adapted features.

### 4.1    Positive Alignment via Prototype Regression

We begin with the ELM formulation, where ridge regression is employed to map the features from the RP layer to the target vectors, as previously defined:

$$\mathbf{W} = (\mathbf{Z}^\top \mathbf{Z} + \lambda \mathbf{I})^{-1} \mathbf{Z}^\top \mathbf{T}, \tag{7}$$

where $\mathbf{Z} \in \mathbb{R}^{N \times D}$ represents the random feature obtained from the RP layer, and $\mathbf{T}$ contains target vectors for inputs. When $\mathbf{T}$ is one-hot encoded for classification, the cross matrix $\mathbf{Z}^\top \mathbf{T}$ reduces to a matrix where each column is the sum of random features belonging to a given class. This can be factorized as:

$$\mathbf{H} = \mathbf{Z}^\top \mathbf{T} = \mathbf{M} \mathbf{N} \tag{8}$$

Here, $\mathbf{M} \in \mathbb{R}^{D \times C}$ contains the **class prototypes** $\mathbf{m}_c$, defined as the means of the random features of class $c$, and $\mathbf{N} = \mathrm{diag}(n_1, \ldots, n_C)$ holds the number of samples per class. Both $\mathbf{M}$ and $\mathbf{N}$ can be computed incrementally as each task arrives.

This formulation can be extended to arbitrary targets. Instead of using one-hot vectors, we can assign each class a **target prototype**, representing its ideal position in the projected space. These target prototypes can have any dimensionality; however, to preserve the geometric structure of the representations, we set their dimension to match that of the input layer, $d$. Let $\mathbf{P} \in \mathbb{R}^{C \times d}$ denote the target prototype matrix, where each row $\mathbf{p}_c$ specifies the desired projection for class $c$. For instance,

$\mathbf{p}_c$ can be set as the original class mean of the input layer, pulling the feature representations of class $c$ toward their respective mean. In this case, the cross matrix becomes:

$$\mathbf{H} = \mathbf{Z}^\top \mathbf{T} = \sum_c \left( \sum_{i \in c} \mathbf{z}_i \right) \mathbf{p}_c^\top = \sum_c \mathbf{m}_c \mathbf{n}_c \mathbf{p}_c^\top = \mathbf{MNP} \tag{9}$$

This decomposition enables efficient incremental updates by maintaining class prototypes $\mathbf{M}$ computed based on the features from the random projection layer, sample counts $\mathbf{N}$, and target prototypes $\mathbf{P}$. The learned projection $\mathbf{W}$ maps samples toward their corresponding target prototypes, effectively pulling intra-class samples together and enhancing positive alignment.

## 4.2 Negative Repulsion via Target-Prototype Separation

Aligning input samples to their respective class means (using class means as target prototypes) improves intra-class compactness but does not guarantee adequate inter-class separation. In particular, some class means may lie close to each other in the feature space, making them hard to distinguish. To address this, we refine the target prototypes by **shifting the class means** to enhance the separation between classes.

Let $\mathbf{C} = (\mu_1, \ldots, \mu_C) \in \mathbb{R}^{d \times C}$ denote the matrix of class means at the input layer. We normalize these means by whitening them with respect to the feature covariance matrix $\mathbf{\Sigma}$ (shared by all classes). This adjustment is essential because a large variance in some dimensions can give a false impression of class separability. For example, in a low-variance dimension even a small separation is meaningful, whereas in high-variance dimensions, a much larger separation is needed to achieve the same effect. Whitening corrects for this imbalance by accounting for the variance of each feature dimension. The covariance is computed incrementally as [15]:

$$\mathbf{\Sigma}_t = \frac{N_{t-1}}{N_t} \mathbf{\Sigma}_{t-1} + \frac{1}{N_t} \sum_{c \in \mathcal{C}_t} \sum_{i \in c} (\mathbf{x}_i - \boldsymbol{\mu}_c)(\mathbf{x}_i - \boldsymbol{\mu}_c)^\top, \tag{10}$$

where $N_t$ is the total number of samples seen up to task $t$ and $\mathcal{C}_t$ is the set of classes introduced at task $t$. We can then transform the class means as:

$$\hat{\mathbf{C}} = \mathbf{\Sigma}^{-1/2} \mathbf{C}, \tag{11}$$

yielding whitened means with unit variance in all directions. We then perform singular value decomposition (SVD) on the whitened class means:

$$\hat{\mathbf{C}} = \mathbf{USV}^\top, \tag{12}$$

where $\mathbf{U} \in \mathbb{R}^{d \times C}$ is an orthonormal basis of the input space, $\mathbf{V} \in \mathbb{R}^{C \times C}$ defines the class means subspace, and $\mathbf{S} \in \mathbb{R}^{C \times C}$ is a diagonal matrix containing the singular values.

Our objective is to enhance the separation between class means, quantified by the sum of their pairwise cosine similarities. In principle, maximum separation would be achieved if the means were arranged uniformly on the surface of a hypersphere. However, this strict configuration would significantly alter the original data geometry, distorting the feature representations. To maintain the structure while increasing class separation, we introduce an adjustment to the singular values in $\mathbf{S}$, which perturbs the class means in directions that reduce their pairwise cosine similarities:

$$\tilde{\mathbf{S}} = \mathbf{S} + \alpha Diag\{\delta_1, \ldots, \delta_C\}, \tag{13}$$

where $\alpha$ is a scaling factor, and $\delta_i$ is defined in the following Lemma.

**Lemma 4.1.** *Denote* $w_1, \ldots, w_C \in \mathbb{R}^C$ *as arbitrary vectors, and* $e_1, \ldots, e_C \in \mathbb{R}^C$ *as a set of orthogonal bases of* $\mathbb{R}^C$. *Denote* $\langle x, y \rangle = x^\top y$ *as the inner product and* $\theta(x, y) = \frac{x^\top y}{||x|| \cdot ||y||}$ *as the cosine similarity. There exist* $\alpha > 0$ *and*

$$\delta_i = \begin{cases} 1, \text{ if } \sum_{j \neq i} \frac{1}{||w_j||} \left[ (2 \cdot 1_{\langle w_i, w_j \rangle \geq 0}) \cdot \langle e_i, w_j \rangle \cdot ||w_i||^2 - \langle e_i, w_i \rangle \cdot |\langle w_i, w_j \rangle| \right] < 0, \\ 0, \text{ if } \sum_{j \neq i} \frac{1}{||w_j||} \left[ (2 \cdot 1_{\langle w_i, w_j \rangle \geq 0}) \cdot \langle e_i, w_j \rangle \cdot ||w_i||^2 - \langle e_i, w_i \rangle \cdot |\langle w_i, w_j \rangle| \right] = 0, \\ -1, \text{ else,} \end{cases}$$

$$\sum_i \sum_{j \neq i} |\theta(w_i + \alpha \delta_i e_i, w_j + \alpha \delta_j e_j)| \leq \sum_i \sum_{j \neq i} |\theta(w_i, w_j)| \tag{14}$$

See the proof in Appendix A. In Eq. 12, we denote $\mathbf{SV}^\top = (w_1, \ldots, w_C)$ and $\mathbf{V}^\top = (e_1, \ldots, e_C)$. Each $w_i$ in the subspace $\mathbf{SV}^\top$ is perturbed by $\alpha \delta_i e_i$, effectively shifting it in an orthogonal direction with magnitude $\alpha$. By Lemma 4.1, there exist coefficients $\delta_1, \ldots, \delta_C$ and a scalar $\alpha > 0$ such that Eq. 14 holds. The sign of each $\delta_i$ determines whether the shift occurs in the direction of $e_i$ or its opposite. This condition arises from the proof of Lemma 4.1, where we analyze the derivative of the function $f_i(\alpha)$, defined as the sum of cosine similarities involving class $i$. By selecting $\delta_i$ accordingly, we ensure that the slope of $f_i(\alpha)$ at $\alpha = 0$ is negative, which guarantees that a small positive $\alpha$ decreases the cosine similarity. The choice of $\alpha$ is also important: if $\alpha$ is too small, the shift becomes negligible, whereas if it is too large, the reduction in cosine similarity may no longer be valid. In practice, we set $\alpha = 1$ in our experiments.

Defining $\mathbf{\Delta} = Diag\{\delta_1, \ldots, \delta_C\}$, the lemma guarantees that the average cosine similarity among the columns of $(\mathbf{S} + \alpha \mathbf{\Delta})\mathbf{V}^\top$ is smaller than that of $\mathbf{SV}^\top$. Since the columns of $\mathbf{U} \in \mathbb{R}^{d \times C}$ form an orthogonal basis in $\mathbb{R}^d$, the transformations preserve inner products and norms:

$$\langle \mathbf{U}w_i, \mathbf{U}w_j \rangle = \langle w_i, w_j \rangle, \quad ||\mathbf{U}w_i|| = ||w_i||, \quad ||\mathbf{U}w_j|| = ||w_j|| \tag{15}$$

Thus, the cosine similarity between transformed vectors remains the same, i.e., $\theta(\mathbf{U}w_i, \mathbf{U}w_j) = \theta(w_i, w_j)$. Therefore, the average cosine similarity among the columns of $\mathbf{U}(\mathbf{S} + \alpha \mathbf{\Delta})\mathbf{V}^\top$ is also reduced compared to the original matrix $\hat{\mathbf{C}} = \mathbf{USV}^\top$.

Note that separability is measured as the sum of cosine similarities, but the class means are whitened. This procedure is closely related to Mahalanobis distance, which has been shown to outperform cosine similarity in related contexts [15]. By whitening, we effectively assess separation using Mahalanobis distance; after increasing the separation, we de-whiten the features to restore their original scale:

$$\hat{\tilde{\mathbf{C}}} = \mathbf{U}\tilde{\mathbf{S}}\mathbf{V}^\top, \quad \tilde{\mathbf{C}} = \hat{\tilde{\mathbf{C}}}\mathbf{\Sigma}^{1/2} \tag{16}$$

The separated class means are then used as target prototypes for the CP layer by setting $\mathbf{P}$ in Eq. 9 to $\tilde{\mathbf{C}}$, aligning samples with their respective classes while enhancing the separation between classes.

### 4.3 Classifier

The CP layer is an analytic module that adapts feature representations across tasks without modifying the underlying PTM. To increase its capacity, we extend it with multiple random projections. Since each RP is initialized independently, we learn a corresponding CP for each RP, all sharing the same set of target prototypes. Specifically, we define $H$ such heads, where each CP head $\mathbf{W}^{(h)}$ maps the random features to the same target prototype space. Given an input $\mathbf{x}$, each head produces a projection and the output is obtained by averaging across heads:

$$\mathbf{u}^{(h)} = \phi(\mathbf{x}\mathbf{R}^{(h)})\mathbf{W}^{(h)}, \quad \mathbf{u} = \frac{1}{H}\sum_{h=1}^{H} \mathbf{u}^{(h)} \tag{17}$$

This aggregated representation $\mathbf{u}$ is then classified using NCM, which assigns it to the nearest target prototype in $\mathbf{P}$. As shown in Table 3 (row 3), AnaCP with an NCM classifier already outperforms all baselines. However, NCM is often surpassed by more expressive classifiers. To further improve performance, we place an ELM classifier after the averaged CP representation.

Unlike PTM features and their random projections, which remain fixed, the CP outputs evolve as new tasks are introduced due to the incremental computation of the CP layer. Directly updating the ELM classifier incrementally (Eq. 3) may therefore lead to CF. To mitigate this, we employ a pseudo-replay strategy. We model the PTM features (input layer) as a multivariate Gaussian distribution parameterized by the class means and a shared covariance matrix, from which we generate pseudo-replay samples for training. This approach resembles that of [10], which also trains the classifier on generated features. However, whereas [10] maintains a separate covariance matrix per class (leading to memory cost that grows with the number of classes), we use a single shared covariance matrix across all classes. This design significantly reduces memory usage while yielding comparable

performance. Specifically, we reuse the class means and shared covariance matrix from Eq. 10 for feature generation. The generated samples are then processed by the RP and CP layers to form the classifier input, upon which the ELM is trained using Eq. 6.

# 5  Experiments

## 5.1  Experimental Setup

**Datasets:** We conduct experiments on five publicly available datasets: CIFAR100 (100 classes) [61], ImageNet-R (200 classes) [62], CUB (200 classes) [63], TinyImageNet (200 classes) [64], and Cars (196 classes) [65]. For all datasets, we use the official train and test splits. Each dataset is split into 10 disjoint tasks by shuffling classes, and experiments are repeated with three different random seeds to account for variability in class-task assignments.

**Baselines:** We compare AnaCP with a range of baselines, including prompt-learning methods (L2P [11], DualPrompt [47], and CODA-Prompt [50]), fine-tuning method (SLCA [10]), and analytic methods that treat the PTM as a frozen feature extractor. The analytic baselines include SimpleCIL [55], SLDA [57], KLDA [14], GACL [54], APER [56], FeCAM [15], and RanPac [13], with some incorporating FSA. For a complete performance perspective, we also include results from joint linear probe, where a linear softmax classifier is trained on the PTM features using gradient descent with all tasks' data, and joint fine-tuning, which directly fine-tunes the PTM on the entire dataset and serves as the upper bound for CIL performance.

**Implementation Details:** For the PTMs, we use DINO-v2 [66] and MoCo-v3 [67], both of which are self-supervised PTMs. This choice helps prevent information leakage, as supervised PTMs are exposed to class labels during training, some of which may reappear in CIL, leading to unintended information leakage. MoCo-v3 is pre-trained on ImageNet-1k – a commonly used but relatively small dataset for pre-training in prior studies. However, because ImageNet-1k (or even ImageNet-21k) is considered limited for real-world applications, we also utilize DINO-v2, a more powerful model pre-trained on the substantially larger LVD-142M dataset. For joint fine-tuning, we employ LoRA adapters [68] instead of full fine-tuning, as we found this approach to be more accurate.

The prompt learning baselines are taken from the [50] repository, while SLCA, FeCAM, and RanPac are evaluated using their original implementations. All remaining analytic baselines are incorporated into a unified experimental setup, following their official implementations to ensure fair comparison.[1] We also adopt FSA following [56] before applying AnaCP to the frozen PTM as it helps improve accuracy. We use RP dimension $D = 5000$, number of CP heads $H = 3$, and number of generated feature representations per class for the classifier $R = 100$ as the default configuration; ablations on other variants are included. The regularization parameter $\lambda$ in Eq. 6 for ELM is set to $10^2$, and the coefficient $\alpha$ for class mean repulsion is set to 1. These values were optimized on a validation set derived from the CIFAR100 training set and are consistently used across all datasets and both PTMs. All experiments are conducted on a single NVIDIA A100 GPU with 80GB VRAM.

**Evaluation Metric:** We evaluate model performance using two primary metrics: Last Accuracy ($A_{\text{Last}}$), the accuracy after completing all tasks, and Average Incremental Accuracy ($A_{\text{Avg}}$), calculated as $A_{\text{Avg}} = \frac{1}{T} \sum_{t=1}^{T} A_t$, where $A_t$ denotes the accuracy at the end of task $t$. Additionally, we measure running time and memory efficiency to assess the practicality of the methods. We also define relative error reduction as $\frac{A_I - A_0}{100 - A_0} \times 100$, where $A_0$ is the baseline accuracy and $A_I$ is the improved accuracy.

## 5.2  Comparison with Baselines

Table 1 presents the main results of our experiments, with all methods using DINO-v2 as the PTM. AnaCP consistently outperforms all baselines across the evaluated datasets, achieving an absolute improvement in last accuracy ranging from 1.03% to 3.17% across different datasets, corresponding to a relative error reduction between 4.8% and 31.4%. We can observe that using replay features with a shared covariance (AnaCP) achieves similar results to class-specific covariances (AnaCP - $\Sigma_y$) while significantly reducing the number of parameters. AnaCP also surpasses the joint linear probe on all datasets and achieves comparable accuracy to joint fine-tuning, which represents the

---

[1]The code of AnaCP is available at https://github.com/SalehMomeni/AnaCP.

| Method | CIFAR100 | | ImageNet-R | | CUB | | TinyImageNet | | Cars | |
|---|---|---|---|---|---|---|---|---|---|---|
| | $A_{avg}$ | $A_{last}$ | $A_{avg}$ | $A_{last}$ | $A_{avg}$ | $A_{last}$ | $A_{avg}$ | $A_{last}$ | $A_{avg}$ | $A_{last}$ |
| Joint linear probe | - | $89.29_{\pm0.00}$ | - | $82.49_{\pm0.02}$ | - | $89.33_{\pm0.01}$ | - | $85.43_{\pm0.01}$ | - | $86.84_{\pm0.01}$ |
| Joint fine-tuning | - | $93.35_{\pm0.00}$ | - | $88.79_{\pm0.01}$ | - | $89.87_{\pm0.01}$ | - | $88.81_{\pm0.02}$ | - | $89.92_{\pm0.02}$ |
| L2P | $88.50_{\pm0.38}$ | $84.16_{\pm0.35}$ | $84.77_{\pm0.64}$ | $79.57_{\pm0.13}$ | $82.48_{\pm1.25}$ | $72.45_{\pm0.25}$ | $89.39_{\pm0.18}$ | $84.45_{\pm0.45}$ | $54.29_{\pm2.02}$ | $40.83_{\pm1.12}$ |
| DualPrompt | $88.13_{\pm0.35}$ | $83.46_{\pm0.14}$ | $84.59_{\pm0.83}$ | $79.61_{\pm0.12}$ | $82.13_{\pm1.25}$ | $72.33_{\pm0.10}$ | $89.50_{\pm0.01}$ | $84.64_{\pm0.26}$ | $53.31_{\pm2.83}$ | $39.15_{\pm1.18}$ |
| CODA-Prompt | $90.87_{\pm0.39}$ | $86.83_{\pm0.01}$ | $86.43_{\pm0.61}$ | $82.39_{\pm0.01}$ | $83.11_{\pm1.07}$ | $73.08_{\pm0.25}$ | $90.58_{\pm0.07}$ | $85.50_{\pm0.52}$ | $58.49_{\pm2.32}$ | $42.63_{\pm0.71}$ |
| SLCA | $93.03_{\pm0.32}$ | $88.80_{\pm0.22}$ | $\underline{88.50}_{\pm0.51}$ | $\underline{84.78}_{\pm0.26}$ | $92.83_{\pm0.52}$ | $88.97_{\pm0.16}$ | $89.38_{\pm0.31}$ | $85.31_{\pm0.21}$ | $90.70_{\pm0.64}$ | $85.76_{\pm0.49}$ |
| SimpleCIL | $91.18_{\pm0.18}$ | $86.20_{\pm0.00}$ | $80.94_{\pm0.46}$ | $75.11_{\pm0.00}$ | $91.20_{\pm0.48}$ | $86.87_{\pm0.00}$ | $87.80_{\pm0.56}$ | $83.17_{\pm0.00}$ | $66.75_{\pm0.39}$ | $55.36_{\pm0.00}$ |
| SLDA | $92.28_{\pm0.31}$ | $87.57_{\pm0.00}$ | $83.28_{\pm0.14}$ | $77.25_{\pm0.00}$ | $93.05_{\pm0.44}$ | $89.51_{\pm0.00}$ | $87.86_{\pm0.69}$ | $82.69_{\pm0.00}$ | $89.97_{\pm0.70}$ | $87.20_{\pm0.00}$ |
| KLDA | $92.97_{\pm0.20}$ | $88.64_{\pm0.16}$ | $82.37_{\pm0.19}$ | $77.75_{\pm0.01}$ | $93.11_{\pm0.33}$ | $\underline{89.54}_{\pm0.11}$ | $88.20_{\pm0.64}$ | $82.91_{\pm0.08}$ | $91.30_{\pm0.63}$ | $86.40_{\pm0.03}$ |
| GACL | $93.85_{\pm0.21}$ | $\underline{90.10}_{\pm0.01}$ | $84.91_{\pm0.23}$ | $81.74_{\pm0.15}$ | $\underline{93.22}_{\pm0.47}$ | $89.37_{\pm0.10}$ | $\underline{90.69}_{\pm0.42}$ | $\underline{86.70}_{\pm0.03}$ | $89.76_{\pm0.48}$ | $84.52_{\pm0.18}$ |
| APER | $93.05_{\pm0.43}$ | $88.59_{\pm0.14}$ | $86.32_{\pm0.12}$ | $80.61_{\pm0.04}$ | $91.18_{\pm0.51}$ | $86.85_{\pm0.12}$ | $89.26_{\pm0.65}$ | $84.50_{\pm0.21}$ | $75.40_{\pm0.97}$ | $65.11_{\pm0.80}$ |
| FeCAM | $93.27_{\pm0.38}$ | $89.08_{\pm0.70}$ | $84.61_{\pm0.46}$ | $79.27_{\pm0.50}$ | $91.59_{\pm0.72}$ | $87.63_{\pm0.53}$ | $89.47_{\pm0.50}$ | $84.75_{\pm0.25}$ | $90.64_{\pm0.98}$ | $85.72_{\pm0.72}$ |
| RanPAC | $\underline{94.10}_{\pm0.34}$ | $90.09_{\pm0.79}$ | $87.96_{\pm0.19}$ | $84.41_{\pm0.20}$ | $92.37_{\pm0.68}$ | $88.44_{\pm0.37}$ | $89.55_{\pm0.48}$ | $84.67_{\pm1.04}$ | $\underline{91.48}_{\pm0.87}$ | $\underline{87.48}_{\pm0.11}$ |
| AnaCP - $\Sigma_y$ | $95.43_{\pm0.18}$ | $92.15_{\pm0.11}$ | $90.38_{\pm0.09}$ | $86.68_{\pm0.03}$ | $93.82_{\pm0.28}$ | $90.61_{\pm0.13}$ | $91.81_{\pm0.26}$ | $87.71_{\pm0.29}$ | $94.29_{\pm0.59}$ | $90.87_{\pm0.20}$ |
| AnaCP | $\mathbf{95.43}_{\pm0.20}$ | $\mathbf{92.15}_{\pm0.09}$ | $\mathbf{90.37}_{\pm0.11}$ | $\mathbf{86.60}_{\pm0.04}$ | $\mathbf{93.84}_{\pm0.31}$ | $\mathbf{90.57}_{\pm0.15}$ | $\mathbf{91.57}_{\pm0.29}$ | $\mathbf{87.35}_{\pm0.29}$ | $\mathbf{94.16}_{\pm0.61}$ | $\mathbf{90.65}_{\pm0.24}$ |
| Rel. Error | 22.5% ↓ | 20.7% ↓ | 16.2% ↓ | 11.9% ↓ | 9.1% ↓ | 9.8% ↓ | 9.4% ↓ | 4.8% ↓ | 31.4% ↓ | 25.3% ↓ |

Table 1: Comparison of AnaCP with baselines using **DINO-v2** as the PTM. All methods are replay-free. AnaCP employs a shared covariance matrix for pseudo-replay feature generation, while AnaCP-$\Sigma_y$ uses a separate covariance matrix for each class; the latter is reported only as an ablation since it requires more parameters. The best result (excluding AnaCP-$\Sigma_y$) is shown in bold, and the second-best is underlined. Rel. Error indicates the reduction in error rate achieved by AnaCP compared to the best baseline for the column.

| Method | CIFAR100 | | ImageNet-R | | CUB | | TinyImageNet | | Cars | |
|---|---|---|---|---|---|---|---|---|---|---|
| | $A_{avg}$ | $A_{last}$ | $A_{avg}$ | $A_{last}$ | $A_{avg}$ | $A_{last}$ | $A_{avg}$ | $A_{last}$ | $A_{avg}$ | $A_{last}$ |
| Joint linear probe | - | $83.81_{\pm0.01}$ | - | $59.16_{\pm0.02}$ | - | $63.19_{\pm0.02}$ | - | $81.12_{\pm0.02}$ | - | $41.95_{\pm0.02}$ |
| Joint fine-tuning | - | $87.45_{\pm0.02}$ | - | $73.54_{\pm0.03}$ | - | $79.24_{\pm0.01}$ | - | $82.54_{\pm0.03}$ | - | $81.98_{\pm0.02}$ |
| SLCA | $69.54_{\pm0.58}$ | $69.54_{\pm0.89}$ | $56.66_{\pm1.12}$ | $44.63_{\pm0.76}$ | $74.42_{\pm0.51}$ | $70.88_{\pm0.36}$ | $68.93_{\pm0.59}$ | $63.53_{\pm0.26}$ | $55.47_{\pm0.88}$ | $53.00_{\pm0.41}$ |
| SimpleCIL | $79.27_{\pm0.47}$ | $70.89_{\pm0.00}$ | $45.83_{\pm0.83}$ | $37.75_{\pm0.00}$ | $56.36_{\pm0.61}$ | $45.83_{\pm0.00}$ | $77.31_{\pm0.46}$ | $71.01_{\pm0.00}$ | $28.99_{\pm0.36}$ | $19.79_{\pm0.00}$ |
| SLDA | $86.68_{\pm0.51}$ | $79.14_{\pm0.00}$ | $65.37_{\pm1.08}$ | $58.22_{\pm0.00}$ | $\underline{79.61}_{\pm0.60}$ | $72.39_{\pm0.00}$ | $81.60_{\pm0.78}$ | $75.30_{\pm0.00}$ | $78.33_{\pm0.32}$ | $69.72_{\pm0.00}$ |
| KLDA | $88.27_{\pm0.29}$ | $81.15_{\pm0.18}$ | $63.64_{\pm0.87}$ | $56.92_{\pm0.27}$ | $78.49_{\pm0.27}$ | $71.39_{\pm0.14}$ | $82.46_{\pm0.72}$ | $75.67_{\pm0.05}$ | $77.33_{\pm0.20}$ | $69.11_{\pm0.09}$ |
| GACL | $\underline{88.89}_{\pm0.31}$ | $\underline{82.10}_{\pm0.10}$ | $64.02_{\pm1.19}$ | $56.80_{\pm0.48}$ | $76.12_{\pm0.32}$ | $67.43_{\pm0.35}$ | $\underline{84.72}_{\pm0.58}$ | $\underline{78.78}_{\pm0.10}$ | $72.48_{\pm0.45}$ | $61.55_{\pm0.26}$ |
| APER | $80.51_{\pm1.25}$ | $71.36_{\pm1.36}$ | $64.65_{\pm0.18}$ | $54.79_{\pm0.68}$ | $72.09_{\pm1.59}$ | $63.34_{\pm0.57}$ | $80.06_{\pm0.74}$ | $72.84_{\pm0.31}$ | $45.09_{\pm2.78}$ | $32.65_{\pm2.06}$ |
| FeCAM | $84.73_{\pm1.24}$ | $77.23_{\pm2.00}$ | $64.61_{\pm0.70}$ | $55.45_{\pm0.69}$ | $73.61_{\pm0.80}$ | $66.50_{\pm0.13}$ | $81.28_{\pm1.44}$ | $74.49_{\pm1.04}$ | $74.55_{\pm0.64}$ | $66.61_{\pm0.87}$ |
| RanPAC | $86.99_{\pm1.46}$ | $79.37_{\pm1.84}$ | $\underline{70.68}_{\pm0.17}$ | $\underline{62.50}_{\pm0.81}$ | $78.93_{\pm0.75}$ | $\underline{72.69}_{\pm0.37}$ | $82.97_{\pm1.01}$ | $76.25_{\pm0.60}$ | $\underline{80.39}_{\pm0.23}$ | $\underline{72.61}_{\pm0.65}$ |
| AnaCP - $\Sigma_y$ | $90.16_{\pm0.36}$ | $83.87_{\pm0.13}$ | $74.88_{\pm0.48}$ | $67.91_{\pm0.48}$ | $84.37_{\pm0.73}$ | $79.05_{\pm0.63}$ | $85.88_{\pm0.58}$ | $80.02_{\pm0.32}$ | $85.08_{\pm0.94}$ | $79.24_{\pm0.63}$ |
| AnaCP | $\mathbf{89.91}_{\pm0.43}$ | $\mathbf{83.70}_{\pm0.26}$ | $\mathbf{74.60}_{\pm0.54}$ | $\mathbf{67.30}_{\pm0.48}$ | $\mathbf{84.21}_{\pm0.76}$ | $\mathbf{78.61}_{\pm0.47}$ | $\mathbf{85.43}_{\pm0.49}$ | $\mathbf{79.32}_{\pm0.51}$ | $\mathbf{84.78}_{\pm0.87}$ | $\mathbf{78.70}_{\pm0.76}$ |
| Rel. Error | 9.1% ↓ | 8.9% ↓ | 13.3% ↓ | 12.7% ↓ | 22.5% ↓ | 21.6% ↓ | 4.6% ↓ | 2.54% ↓ | 22.3% ↓ | 22.2% ↓ |

Table 2: Comparison of AnaCP with baselines using **MoCo-v3** as the PTM. The prompt learning baselines are excluded here due to their much lower accuracy, which is also the case in Table 1.

upper bound for CIL. Notably, on two datasets, AnaCP even exceeds this upper bound, while on the remaining datasets, the largest accuracy gap is only around 2%. These results highlight that AnaCP's feature adaptation is highly effective, even without directly training the PTM on each task.

We also evaluate AnaCP using MoCo-v3 as the PTM, as presented in Table 2. AnaCP outperforms the baselines by a larger margin with MoCo-v3 than with DINO-v2, as the weaker features of MoCo-v3 make the CP layer more impactful. In this case, the gap to joint fine-tuning increases to 6%, which is expected since MoCo-v3 is a weaker PTM and benefits more from full fine-tuning. However, given that both PTMs have the same architecture (number of layers and hidden size), there is no reason to prefer a weaker PTM in real-world applications.

## 5.3 Analysis of Catastrophic Forgetting

The CP layer in our method is inherently immune to CF, as it operates on a frozen PTM and updates its statistics (prototypes and the Gram matrix) incrementally without overwriting prior information. In contrast, the final ELM classifier relies on pseudo-replay, which in principle could introduce some forgetting. In practice, however, we found this effect to be minimal. To verify this, we evaluate under the task-incremental learning (TIL) setting, where the task identity is provided at inference time. Specifically, we present the accuracy matrix $A[t][i]$, where each entry denotes the accuracy on task $i$ after training on the first $t$ tasks of CIFAR-100 (10-task split) using DINO-v2 as the backbone.

We adopt the TIL setting for this analysis because it offers a clearer view of forgetting. In CIL, accuracy inevitably decreases as more tasks are introduced, not necessarily due to forgetting, but because the classification problem becomes harder with a growing number of classes. By contrast, TIL keeps the classification difficulty fixed, since each task has a constant number of classes and the

Figure 2: Accuracy matrix $A[t][i]$ on CIFAR-100 with 10 task splits using DINO-v2 as the backbone under the TIL setting. Each entry shows the accuracy on task $i$ after training on the first $t$ tasks.

task ID is known at inference. Under this setup, forgetting would appear as a decline in accuracy on earlier tasks. However, as shown in Figure 2, the accuracy for task 1 (first column) remains nearly constant across all steps. The same holds for other tasks, demonstrating that our method effectively preserves knowledge from prior tasks.

| row | CLS | CP | NR | $H$ | $D$ | $R$ | CIFAR100 | ImageNet-R | CUB | TinyImageNet | Cars |
|---|---|---|---|---|---|---|---|---|---|---|---|
| 1 | NCM | ✗ | - | - | - | - | $88.13_{\pm 0.21}$ | $81.60_{\pm 0.08}$ | $86.95_{\pm 0.36}$ | $83.63_{\pm 0.48}$ | $69.51_{\pm 0.87}$ |
| 2 | NCM | ✓ | ✗ | 3 | 5000 | - | $91.31_{\pm 0.03}$ | $83.29_{\pm 0.45}$ | $89.52_{\pm 0.10}$ | $85.89_{\pm 0.26}$ | $86.25_{\pm 0.25}$ |
| 3 | NCM | ✓ | ✓ | 3 | 5000 | - | $91.86_{\pm 0.05}$ | $86.01_{\pm 0.11}$ | $89.88_{\pm 0.32}$ | $87.02_{\pm 0.27}$ | $89.48_{\pm 0.29}$ |
| 4 | ELM | ✗ | - | - | 5000 | - | $91.12_{\pm 0.10}$ | $85.14_{\pm 0.26}$ | $89.75_{\pm 0.19}$ | $86.48_{\pm 0.12}$ | $89.32_{\pm 0.20}$ |
| 5 | ELM | ✓ | ✗ | 3 | 5000 | 100 | $91.87_{\pm 0.02}$ | $85.73_{\pm 0.19}$ | $90.39_{\pm 0.14}$ | $86.69_{\pm 0.28}$ | $90.24_{\pm 0.20}$ |
| 6 | ELM | ✓ | ✓ | 3 | 5000 | 100 | $92.15_{\pm 0.09}$ | $86.60_{\pm 0.04}$ | $90.48_{\pm 0.15}$ | $87.71_{\pm 0.29}$ | $90.65_{\pm 0.24}$ |
| 7 | ELM | ✓ | ✓ | 3 | 5000 | 20 | $92.14_{\pm 0.06}$ | $86.39_{\pm 0.04}$ | $90.43_{\pm 0.18}$ | $87.29_{\pm 0.32}$ | $90.45_{\pm 0.23}$ |
| 8 | ELM | ✓ | ✓ | 3 | 5000 | 50 | $92.13_{\pm 0.04}$ | $86.47_{\pm 0.05}$ | $90.46_{\pm 0.10}$ | $87.49_{\pm 0.32}$ | $90.57_{\pm 0.16}$ |
| 9 | ELM | ✓ | ✓ | 3 | 5000 | 100 | $92.15_{\pm 0.09}$ | $86.60_{\pm 0.04}$ | $90.48_{\pm 0.15}$ | $87.71_{\pm 0.29}$ | $90.65_{\pm 0.24}$ |
| 10 | ELM | ✓ | ✓ | 1 | 5000 | 100 | $91.99_{\pm 0.01}$ | $85.72_{\pm 0.24}$ | $90.36_{\pm 0.13}$ | $87.12_{\pm 0.24}$ | $90.35_{\pm 0.24}$ |
| 11 | ELM | ✓ | ✓ | 3 | 5000 | 100 | $92.15_{\pm 0.09}$ | $86.60_{\pm 0.04}$ | $90.48_{\pm 0.15}$ | $87.71_{\pm 0.29}$ | $90.65_{\pm 0.24}$ |
| 12 | ELM | ✓ | ✓ | 5 | 5000 | 100 | $92.20_{\pm 0.04}$ | $86.62_{\pm 0.10}$ | $90.51_{\pm 0.05}$ | $87.76_{\pm 0.29}$ | $90.68_{\pm 0.18}$ |
| 13 | ELM | ✓ | ✓ | 3 | 1000 | 100 | $90.64_{\pm 0.10}$ | $84.01_{\pm 0.29}$ | $89.56_{\pm 0.15}$ | $85.48_{\pm 0.17}$ | $88.23_{\pm 0.21}$ |
| 14 | ELM | ✓ | ✓ | 3 | 2000 | 100 | $91.35_{\pm 0.08}$ | $85.15_{\pm 0.19}$ | $89.97_{\pm 0.18}$ | $86.24_{\pm 0.22}$ | $89.93_{\pm 0.10}$ |
| 15 | ELM | ✓ | ✓ | 3 | 5000 | 100 | $92.15_{\pm 0.09}$ | $86.60_{\pm 0.04}$ | $90.48_{\pm 0.15}$ | $87.71_{\pm 0.29}$ | $90.65_{\pm 0.24}$ |
| 16 | ELM | ✓ | ✓ | 3 | 10000 | 100 | $92.75_{\pm 0.10}$ | $86.86_{\pm 0.10}$ | $90.60_{\pm 0.10}$ | $88.11_{\pm 0.24}$ | $90.73_{\pm 0.16}$ |

Table 3: Ablation studies on AnaCP with DINO-v2 as the PTM. All reported values are $A_{\text{last}}$. CLS indicates the method employed in the classifier (Section 4.3). CP indicates whether the proposed contrastive projection is applied. NR (negative repulsion) shows whether class mean separation is improved through Lemma 4.1; when not used, the original class means serve as target prototypes. The table also includes variations in the RP dimension ($D$), the number of heads in the CP layer ($H$), and the number of feature representations generated per class ($R$).

### 5.4 Ablation Studies

We conduct a series of ablations to evaluate the impact of key components, as shown in Table 3. **Rows 1-3** examine the effect of the CP layer. Removing CP and directly applying an NCM classifier to PTM features results in a significant drop in absolute accuracy between 2.93% to 19.97% (row 1 vs. row 3). This highlights the importance of CP in enhancing feature representations, as illustrated in Figure 1. The effect of negative repulsion (NR), which increases the separation between class means (Section 4.2) is also evident; disabling NR and relying only on positive alignment with the original class means as target prototypes leads to an accuracy reduction from 0.38% to 3.23% (row 2 vs. row 3). This confirms that NR is a critical component for improving class separability.

The role of classifier is explored in **Rows 4-6**, where NCM is replaced by ELM. We find that using an ELM classifier on average improves the performance by 0.66% (row 3 vs. row 6). Although this requires generating feature representations as pseudo-replay for training the ELM classifier, as explained in Section 4.3. An arbitrary number of feature representations can be sampled from the Gaussian distribution with negligible computational overhead, as detailed in the running time analysis. We observe that increasing the number of generated feature representations $R$ improves performance on some datasets (**Rows 7-9**). The effects of various numbers of CP heads $H$ (**Rows 10-12**) and RP dimensions $D$ (**Rows 13-16**) are also explored. Generally, increasing these values improves accuracy,

albeit at the cost of additional parameters. For the main results, we have used $D = 5000$ and $H = 3$. We note that even with $H = 1$, AnaCP outperforms all baselines.

## 5.5 Memory and Running Time Efficiency

AnaCP needs to save $C$ class means, each of size $d$, and a shared covariance matrix of size $d \times d$ for feature whitening and pseudo-replay feature generation at the input layer (Figure 1). While a separate covariance matrix can be maintained for each class, this would result in a significant increase in memory as the number of classes grows. As shown in Table 1, using a shared covariance matrix does not compromise accuracy. Each CP head maintains the following: (1) an RP matrix of size $d \times D$, (2) a $D \times D$ Gram matrix, (3) $C$ class prototypes (or means) in the random feature space, each of size $D$, and (4) a projection matrix $W$ of size $d \times D$. We have $H$ such heads in total. The target prototypes do not need to be stored, as they can be derived from the class means. The classifier layer also maintains an RP matrix of size $d \times D$ and $W$ of size $C \times D$. Unlike the CP layer, this layer doesn't need to store the Gram matrix for incremental updates, as it is computed from generated feature representations after each task.

For a typical configuration with $C = 200$, $d = 768$, $D = 5000$, and $H = 3$, the total number of stored parameters amounts to approximately 106.6 million. The majority of these parameters belong to the $D \times D$ Gram matrices, which does not increase with the number of classes. Also, none of these parameters are trainable, making them suitable for storage in reduced-precision formats such as float8. This would further reduce the total memory usage to approximately 102 MiB.

While AnaCP may have a higher parameter count compared to some other methods, the design is balanced with exceptional computational efficiency. For example, CIFAR100 with the DINO-v2 PTM requires only 9 minutes and 18 seconds for training. 6 minutes and 5 seconds of this time are dedicated to the FSA, and 3 minutes and 8 seconds are spent passing inputs through the PTM to extract features. The core operations of AnaCP, including updating the CP layer and training the classifier with generated features, collectively take only 5 seconds. Thus, despite the larger parameter count, these operations are extremely fast. For comparison, a baseline such as CODA-Prompt that requires task-specific training takes 3 hours and 47 minutes to complete training on CIFAR100, and joint fine-tuning requires 1 hour and 58 minutes.

## 5.6 Discussion

Our results support the proposition that a strong PTM is the key to CL, and that representation-level forgetting is not the primary limitation. Even simple CIL strategies, when paired with a fixed PTM, can achieve near-optimal performance. This view aligns with neuroscience evidence that the human brain maintains stable representations, despite minor drift over time [69, 70]. Cortical activity also preserves a stable similarity structure that supports consistent perception and behavior, even as neural responses gradually shift [71, 70]. The human brain can thus be viewed as an innate PTM refined through evolution, which provides a stable foundation for continual adaptation. Similarly, large-scale pre-training in machine learning resembles this biological evolution and development, producing reusable representations that enable CL without any forgetting. For further discussion, see [72].

# 6 Conclusion

CIL poses a significant challenge in continual learning. Recent analytic methods with PTMs have shown promise. However, since they cannot learn or adapt features obtained from PTMs to suit specific CIL tasks, their performance is still suboptimal. This paper proposed a novel and principled method, called AnaCP, to adapt features extracted from PTMs analytically, which offers a highly efficient and accurate solution for CIL. Empirical evaluations demonstrate that AnaCP outperforms strong baselines, but more importantly, if paired with a strong PTM, its accuracy can be comparable to the joint training upper bound.

**Limitations:** When using a strong PTM like DINO-v2, AnaCP achieves accuracy on par with the joint fine-tuning upper bound. However, with a weaker PTM such as MoCo-v3, AnaCP cannot match joint fine-tuning accuracy due to the critical role of PTM features played in our method. While strong PTMs are readily available today, improving AnaCP's performance with weaker PTMs remains a meaningful goal. Additionally, our work currently focuses CIL. We believe AnaCP can be extended to TIL and domain-incremental learning (DIL) scenarios with appropriate adaptations.

## Acknowledgments

The work of Saleh Momeni and Bing Liu was supported in part by three NSF grants (IIS-2229876, IIS-1910424, and CNS-2225427), and an NVIDIA's Academia Grant, which provides cloud compute via its Saturn Cloud.

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

# A  Technical Appendices

**Lemma A.1** (Lemma 4.1). *Denote $w_1, \ldots, w_C \in \mathbb{R}^C$ as arbitrary vectors, and $e_1, \ldots, e_C \in \mathbb{R}^C$ as a set of orthogonal bases of $\mathbb{R}^C$. Denote $\langle x, y \rangle = x^\top y$ as the inner product and $\theta(x, y) = \frac{x^\top y}{||x|| \cdot ||y||}$ as the cosine similarity. There exist $\alpha > 0$ and*

$$
\delta_i = \begin{cases}
1, \text{if } \sum_{j \neq i} \frac{1}{||w_j||} \left[ (2 \cdot 1_{\langle w_i, w_j \rangle \geq 0}) \cdot \langle e_i, w_j \rangle \cdot ||w_i||^2 - \langle e_i, w_i \rangle \cdot |\langle w_i, w_j \rangle| \right] < 0, \\[3mm]
0, \text{if } \sum_{j \neq i} \frac{1}{||w_j||} \left[ (2 \cdot 1_{\langle w_i, w_j \rangle \geq 0}) \cdot \langle e_i, w_j \rangle \cdot ||w_i||^2 - \langle e_i, w_i \rangle \cdot |\langle w_i, w_j \rangle| \right] = 0, \\[3mm]
-1, \text{else},
\end{cases}
$$

*s.t.*

$$
\sum_i \sum_{j \neq i} |\theta(w_i + \alpha \delta_i e_i, w_j + \alpha \delta_j e_j)| \leq \sum_i \sum_{j \neq i} |\theta(w_i, w_j)|. \tag{18}
$$

*Proof.* Let

$$
f_i(\alpha) = \sum_{j \neq i} \frac{|\langle w_j, w_i + \alpha \delta_i e_i \rangle|}{||w_j|| \cdot ||w_i + \alpha \delta_i e_i||}.
$$

For simplicity, denote

$$
p_{i,j}(\alpha) = |\langle w_i + \alpha \delta_i e_i, w_j + \alpha \delta_j e_j \rangle| = |\langle w_i, w_j \rangle + \alpha \delta_i \langle e_i, w_j \rangle|
$$

and

$$
q_i(\alpha) = ||w_i + \alpha \delta_i e_i|| = \sqrt{\langle w_i + \alpha \delta_i e_i, w_i + \alpha \delta_i e_i \rangle}.
$$

Therefore, we have

$$
f_i(\alpha) = \sum_{j \neq i} \frac{1}{||w_j||} \cdot \frac{p_{i,j}(\alpha)}{q_i(\alpha)}.
$$

Since $\frac{d|x|}{dx} = 2 \cdot 1_{x \geq 0} - 1$, we have

$$
p'_{i,j}(\alpha)|_{\alpha=0} = (2 \cdot 1_{\langle w_i, w_j \rangle \geq 0}) \cdot \delta_i \langle e_i, w_j \rangle.
$$

Since $\langle w_i + \alpha \delta_i e_i, w_i + \alpha \delta_i e_i \rangle = \langle w_i, w_i \rangle + 2 \delta_i \langle e_i, w_i \rangle + \alpha^2$, we have

$$
q'_i(\alpha)|_{\alpha=0} = \frac{1}{2} \frac{2 \delta_i \langle e_i, w_i \rangle + 2\alpha}{\sqrt{\langle w_i + \alpha \delta_i e_i, w_i + \alpha \delta_i e_i \rangle}} \Big|_{\alpha=0} = \frac{\delta_i \langle e_i, w_i \rangle}{||w_i||}
$$

Therefore, we have

$$
\begin{aligned}
f'_i(\alpha)|_{\alpha=0} &= \sum_{j \neq i} \frac{1}{||w_j||} \cdot \frac{p'_{i,j}(\alpha) q_i(\alpha) - p_{i,j}(\alpha) q'_i(\alpha)}{q_i^2(\alpha)} \Big|_{\alpha=0} \\
&= \sum_{j \neq i} \frac{1}{||w_j|| ||w_i||^2} \left[ (2 \cdot 1_{\langle w_i, w_j \rangle \geq 0} - 1) \cdot \delta_i \langle e_i, w_j \rangle \cdot ||w_i|| - |\langle w_i, w_j \rangle| \cdot \frac{\delta_i \langle e_i, w_i \rangle}{||w_i||} \right] \\
&= \frac{\delta_i}{||w_i||^3} \sum_{j \neq i} \frac{1}{||w_j||} \left[ (2 \cdot 1_{\langle w_i, w_j \rangle \geq 0} - 1) \cdot \langle e_i, w_j \rangle \cdot ||w_i||^2 - \langle e_i, w_i \rangle \cdot |\langle w_i, w_j \rangle| \right],
\end{aligned}
$$

which shows that $\delta_i$'s defined above guarantee $f'_i(\alpha)|_{\alpha=0} \leq 0$.

Let

$$
g_j(\alpha; \hat{\alpha}) = \sum_{i \neq j} \frac{|\langle w_j + \alpha \delta_j e_j, w_i + \hat{\alpha} \delta_i e_i \rangle|}{||w_j + \alpha \delta_j e_j|| \cdot ||w_i + \hat{\alpha} \delta_i e_i||}.
$$

Since $f'_i(\alpha)$ is a continuous function, we could take a sufficiently small $\hat{\alpha}$ s.t. $f'_i(\alpha')$ has the same sign when $\alpha' \in [0, \hat{\alpha}]$. Then for $\alpha' \in [0, \hat{\alpha}]$, the $\delta_i$'s defined above also guarantee $g'_j(\alpha; \alpha')|_{\alpha=0} \leq 0$. Choosing $\alpha = \alpha'$ gives the result. $\qquad \square$

