# OpenReview forum: "AnaCP: Toward Upper-Bound Continual Learning via Analytic Contrastive Projection"
_NeurIPS.cc/2025/Conference — NeurIPS 2025 spotlight_

### Official Review · Reviewer_fcMA · 2025-07-02

**Clarity:** 3
**Significance:** 2
**Originality:** 2
**Rating:** 4
**Confidence:** 5

**Summary:**

Analytical Contrastive Projection (ACP) tackles class-incremental continual learning by using a frozen pre-trained model (PTM) with an entirely closed-form feature-adaptation pipeline. It first projects PTM features into a higher-dimensional random space, then applies an analytical contrastive projection that pulls same-class embeddings together and pushes different-class embeddings apart in the closed form solutions type, without any gradient-based training of the PTM.

**Questions:**

I will be happy to update the scores if the authors can effectively address the following questions.
1.	Why are exemplars required despite the strong generalization capabilities of the pre-trained models?
2.	Can you compare the performance with and without random projections in the ablation studies?
3.	How is negative repulsions different from the underlying mechanism of contrastive learning containing both positive and negative data? (SimCLR[1] for example)
4.	Why isn’t forgetting performance compared? This gives better insight into the performance numbers, helping understand the stability-plasticity tradeoff.

[1] A Simple Framework for Contrastive Learning of Visual Representations

**Ethical Concerns:**

["NO or VERY MINOR ethics concerns only"]

**Final Justification:**

Authors has addressed major concerns raised by the reviewer, which is reflected in the final rating.

**Limitations:**

Yes

**Quality:**

3

**Strengths And Weaknesses:**

Strengths –
1. Contrastive projection layer that is trainable in a non-iterative closed loop form.
2. Pseudo replay features to train final ELM classifier, requiring synthetic replay but only through closed-form updates.
3. Extensive experiments conducted with different PTM backbones and datasets.

Weakness –
1. Through the strong generalization capabilities of PTM, why do they still require replay of samples.
2. Banking heavily on a prior work RANPAC[1] and adding the ACP layer on top of the well performing RANPAC.
3. Ablation studies should take into consideration of removing random projections and then comparing the performance.
4. It is unclear how contrastive projections is different from negative-repulsion.
[1] RanPAC: Random Projections and Pre-trained Models for Continual Learning (Mark McDonnell, Dong Gong, Amin Parvaneh, Ehsan Abbasnejad, Anton van den Hengel)

---

> ### Author Rebuttal · Authors · 2025-07-29
>
> We acknowledge the similarity of our method to RanPAC; however, RanPAC directly uses features from the frozen PTM without adapting them to downstream task data. Our motivation was to address this limitation by introducing the contrastive projection, which enhances feature quality before classification.
>
> - **Question 1:**
>   Our method has a contrastive projection (CP) layer that improves the features by pulling together samples from the same class and repelling class means from one another. This projection is learned analytically based on the class means and the Gram matrix of the PTM features, and does not require any exemplars or replay data.
>
>   However, the prediction is performed by an ELM classifier built on the output of the CP layer. In principle, we could also learn the ELM classifier analytically (similar to the CP layer), but this would require storing the class means and Gram matrix of the projected features. Unfortunately, the CP layer changes with each new task — each time a task is added, the projection is updated — so unlike the PTM, the output space of the CP layer is not fixed. Therefore, to compute these statistics in the new space, we rely on generated pseudo-replay data.
>
>   Note that if we remove the ELM classifier and use an NCM classifier directly with the target prototypes, we no longer need the pseudo-replay data. The results of the NCM classifier are slightly worse than those of ELM, but still outperform the baselines, as shown in row 3 of Table 3.
>
> - **Question 2:**
>   Thank you for the suggestion. We have included a comparison without the random projections. The results below show final accuracy using the DINOv2 backbone:
>
>   | proj\_dim     | CIFAR-100 | ImageNet-R | CUB   | T-ImageNet | Cars  |
>   | ------------- | --------- | ---------- | ----- | ---------- | ----- |
>   | no projection | 90.35     | 83.29      | 89.34 | 85.30      | 87.29 |
>   | 1000          | 90.64     | 84.01      | 89.56 | 85.48      | 88.23 |
>   | 2000          | 91.35     | 85.15      | 89.97 | 86.24      | 89.93 |
>   | 5000          | 92.15     | 86.60      | 90.48 | 87.71      | 90.65 |
>   | 10000         | 92.75     | 86.86      | 90.60 | 88.11      | 90.73 |
>
>   We observe that using no projection performs slightly worse than setting "proj_dim=1000". As the projection dimension increases, accuracy also improves. Note that the original feature dimension is "d=768".
>
> - **Question 3:**
>   Conceptually, our approach follows a similar idea as SimCLR and other contrastive learning methods: pulling same-class samples together and pushing different classes apart. The key difference lies in how these objectives are achieved. SimCLR and similar methods rely on gradient-based optimization to learn embeddings using a contrastive loss, which is not well-suited for continual learning due to the incremental nature of learning and catastrophic forgetting from backpropagation. Our motivation is to achieve the same contrastive effects analytically, so we can support incremental updates while avoiding forgetting (also see the forgetting results below).
>
> - **Question 4:**
>   We did not initially report forgetting metrics because our method, like most prototype-based methods, is inherently immune to catastrophic forgetting. This is because we operate on a frozen PTM, and all statistics (class means and Gram matrix) can be incrementally updated without overwriting previous information. Note that even if we had access to all data simultaneously, we would end up with the same class means and Gram matrix, indicating that our approach does not suffer from the traditional forgetting.
>
>   Although this might not hold for the final ELM classifier, which is trained using pseudo-replay, we found its effect to be minimal in practice. This also explains why our method can achieve accuracy on par with joint training. To empirically support this, we have included additional results under the task-incremental learning (TIL) setting, where the task ID is known at inference time. We present the matrix A[t][i], whose elements represent the accuracy on task *i* after learning the first *t* tasks on CIFAR-100 with 10 task splits, using DINOv2 as the backbone.
>
>   The matrix A[t][i] is shown below:
>
>       0.987   -       -       -       -       -       -       -       -       -
>       0.987   0.981   -       -       -       -       -       -       -       -
>       0.989   0.978   0.980   -       -       -       -       -       -       -
>       0.988   0.981   0.979   0.984   -       -       -       -       -       -
>       0.988   0.975   0.981   0.986   0.984   -       -       -       -       -
>       0.987   0.977   0.978   0.982   0.980   0.985   -       -       -       -
>       0.988   0.973   0.978   0.982   0.980   0.984   0.958   -       -       -
>       0.987   0.976   0.976   0.979   0.979   0.983   0.955   0.988   -       -
>       0.985   0.973   0.977   0.979   0.979   0.983   0.957   0.984   0.951   -
>       0.986   0.973   0.977   0.978   0.981   0.983   0.955   0.984   0.947   0.979
>
>   We chose the TIL setting for this analysis because it provides clearer insight into forgetting. In CIL, accuracy will naturally drop as more tasks are learned, not necessarily due to forgetting, but because the classification problem becomes increasingly more difficult (e.g., distinguishing between 100 classes is harder than between 10).
>
>   In contrast, TIL keeps the classification difficulty fixed because the task ID is provided at inference time. For example, task 1 always involves classifying among the same 10 classes, regardless of how many tasks have been learned. If forgetting occurred, it would show as a drop in earlier task accuracy. However, as seen in the matrix, the first column (task 1 accuracy) remains nearly constant across all steps. The same applies to other columns, indicating our method retains prior knowledge effectively.
>
>
> We appreciate your comments. If we have addressed your concerns, we would be grateful if you could consider updating your score.

---

> > ### Author Response · Authors · 2025-08-04
> >
> > Dear Reviewer fcMA,
> >
> > We sincerely appreciate your constructive comments and questions. We would be happy to hear whether our responses addressed your concerns, and if there are any remaining questions we would be glad to provide further details.
> >
> > Thank you for your time and consideration.
> >
> > #### &nbsp;
> > Best regards,
> >
> > Authors

---

### Official Review · Reviewer_yx87 · 2025-07-03

**Clarity:** 3
**Significance:** 3
**Originality:** 3
**Rating:** 4
**Confidence:** 4

**Summary:**

This paper focuses on the class-incremental learning (CIL) and proposes ACP (Analytical Contrastive Projection) to enable incremental feature adaptation while maintaining computational efficiency through analytical classifiers. ACP is implemented in a gradient-free method and achieves strong performance on various CIL benchmarks.

**Questions:**

N/A

**Ethical Concerns:**

["NO or VERY MINOR ethics concerns only"]

**Final Justification:**

Thanks for the authors' reply. All my concerns have been addressed, and I will keep my score unchanged.

**Limitations:**

Yes

**Quality:**

3

**Strengths And Weaknesses:**

Strength
1.	This paper is well-written, and the motivation is clear.
2.	This paper proposes a new way to improve the performance of analytical classifier methods, which are limited by the fixed pre-trained model and linear classifier.
3.	Experiments have demonstrated the effectiveness of the proposed ACP on various datasets and settings.

Weakness
1. Positive Alignment
- Improvement in Method Description Clarity: While the mathematical formulation of the proposed method appears rigorous, the current presentation significantly increases the cost of understanding for readers. Additionally, redefining the left-hand side of Eq. 9 as $ H = Z^\top P$ may improve clarity (only suggestion, depending on the authors).
- Dependence on Pretrained Model: The strategy enhances the RP features by class means of the input layer, aiming to enhance the discriminative power of the analytical classifier. However, this approach is inherently dependent on the quality of the pretrained model (PTM), especially when dealing with cross-domain or fine-grained data. As shown in the experiments, the method primarily benefits from strong self-supervised models such as DINO-v2 and MoCo-v3. Briefly, this strategy does not fundamentally improve the separability of features but rather leverages already well-structured representations.

2. Negative Repulsion
- Lack of Intuitive Explanation for Lemma 4.1: The construction of $\delta_i$ is not clearly explained. What specific information or property does this diagonal matrix encode?.
- Need for Discussion on Relationship with Neural Collapse: It would also be valuable to discuss how the proposed negative repulsion mechanism relates to or differs from the phenomenon of *neural collapse*, which describes the tendency of features within each class to collapse to a fixed simplex structure during training.

Minor:
Please carefully check the language and grammar. For example, in Line 3, “Traditional CIL methods does not use” should be “Traditional CIL methods do not use”.

---

> ### Author Rebuttal · Authors · 2025-07-29
>
> - We appreciate the suggestion regarding the presentation of the mathematical formulation. We will improve the readability in the revised paper.
>
> - We acknowledge that our method relies on PTMs for well-learned representations, which was also mentioned in the limitation section in the paper. However, this is a common characteristic of most modern CIL approaches using PTMs such as prototype-based methods, prompt learning, and even fine-tuning, which similarly require access to high-quality representations from PTMs. Additionally, we believe this dependence is not restrictive, as PTMs are increasingly accessible across different domains and modalities, and their quality is also getting better.
>
> - The motivation behind Lemma 4.1 is to move the class means in directions that reduce their pairwise cosine similarity. Specifically, each $w_i$ in the subspace $SV_T$ is perturbed by $\alpha \delta_i e_i$, where $e$ vectors form an orthonormal basis. This means each $w_i$ is shifted in an orthogonal direction by a magnitude of $\alpha$. The sign of $\delta_i$ determines whether the shift is in the direction of $e_i$ or the opposite. This condition arises from the proof of Lemma 4.1, where we take the derivative of the function $f_i(\alpha)$, representing the sum of cosine similarities for class $i$. By choosing $\delta_i$ in this way, we ensure that the slope of $f_i(\alpha)$ at $\alpha = 0$ is negative, guaranteeing that a small positive $\alpha$ will reduce cosine similarity, i.e., shift the class means away from one another.
>
>
> - We agree that drawing connections to neural collapse provides a valuable perspective, and we will address this in the revision. Indeed, our contrastive projection shares important similarities with this phenomenon. In neural collapse, all samples within a class collapse to their class mean, which aligns with our method, where we explicitly map all samples from a class to their respective class means. Neural collapse also leads to class means forming a simplex equiangular tight frame, i.e., every pair of class means is equally separated. However, unlike neural collapse, our goal is not to enforce equidistant class means. Instead, we seek to preserve the underlying geometry of the features while pushing the class means away from one another, inspired by contrastive learning. Also, neural collapse occurs in the terminal phase of training (TFT) when the loss approaches zero, which makes it difficult to achieve in continual learning due to incremental learning and catastrophic forgetting. Our method minimizes inter-class variation and enhances class separability analytically without fine-tuning the PTM.
>
> - We will also carefully review the manuscript for grammar and language clarity.

---

### Official Review · Reviewer_iEp4 · 2025-07-03

**Clarity:** 3
**Significance:** 2
**Originality:** 3
**Rating:** 4
**Confidence:** 3

**Summary:**

The paper tackles continual class-incremental learning with pretrained models

**Questions:**

- How to guarantee the covariance in Eq 10 is strictly positive definite? This is important as in Eq 11 (which has a typo), its inverse is needed to be computed.

**Ethical Concerns:**

["NO or VERY MINOR ethics concerns only"]

**Final Justification:**

The rebuttal has adequately addressed my concerns.

**Limitations:**

See above.

**Paper Formatting Concerns:**

The paper has no formatting concerns.

**Quality:**

3

**Strengths And Weaknesses:**

Strengths. The paper tackles a relevant problem of interest to the continual learning audience. The manuscript overall is clearly written with good coverage and clear discussion on the literature. The proposed method is clear (though could be better motivated, see below). The experimental results look strong and the paper has a nice set of ablation studies.


Weaknesses. My first major concern is about the comparison regarding RanPAC, one of the most relevant approaches. There are a few implementation differences in the provided RanPAC implementation and its original implementation. RanPAC sets ridge parameter by cross validation and sets dimension $D$ to $10,000$, and uses ReLU as its activation. The paper sets smaller dimension $D=5000$, and  sets a fixed number 100 for both RanPAC and the proposed method, and the proposed method instead uses GELU. It would make more sense and make the paper more convincing if their setups are unified with the same activation (with experiments on both ReLU and GELU), and if RanPAC uses its original dimension and original cross validation approach.

The second major concern is on the technical development. Section 4 reads like a cookbook and the intuition or motivation behind the strategy proposed is often unclear at least to me. More specifically:
- What is the motivation to regress a target prototype which is defined to be the class means? If regressing the class mean is a good goal, why don't we just output the class mean of that layer? Note that we can compute the class mean directly. We want to regress the one-hot label as there is no explicit map from features to labels, but we do have an explicit map from features to their means.
- Why are whitening and de-whitening steps essential? I imagine that we can do SVD directly on C and correct its singular values. It is unclear to me why the whitening is applied on the left in Eq 11 and de-whitening is on the right at Eq 16. Could the authors explain this in more detail?
- I am confused around Eq 13 and Lemma 4.1. What's that intuition behind Eq 13? Namely, why adding a diagonal shift would reduce cosine similarity? Further, how does this condition on $\delta_i$ arise? How large $\alpha$ is, and what is the rationale of choosing $\alpha=1$? Can Lemma 4.1 be proved for $\alpha=1$? In general I find a detailed explanation of Lemma 4.1 on its significance is lacking.



Finally, it would be nice if the authors compare the following baselines either in text or in experiments:
- HidePrompt, which seems to work very well among prompt based methods (https://proceedings.neurips.cc/paper_files/paper/2023/hash/d9f8b5abc8e0926539ecbb492af7b2f1-Abstract-Conference.html).
- LoRanPAC, which seems to be a stronger alternative of RanPAC (https://openreview.net/forum?id=bqv7M0wc4x)
- InfLoRA, which is a representative adapter-based method and the paper seems to compare no methods from this category (https://openaccess.thecvf.com/content/CVPR2024/html/Liang_InfLoRA_Interference-Free_Low-Rank_Adaptation_for_Continual_Learning_CVPR_2024_paper.html)

---

> ### Author Rebuttal · Authors · 2025-07-31
>
> - **Comparison regarding RanPac:**
>
> As shown in Table 3, increasing the dimension $D$ generally leads to improved performance. We wanted to use the same dimension for all the methods that utilize this technique (KLDA also uses $D=5000$). Therefore, we set $D=5000$ for all methods to allow a fair comparison. Regarding the activation function, we did not observe significant differences between ReLU and GELU in our experiments. We opted for GELU as it seemed more intuitive: since the goal is to expand the feature dimension, ReLU would zero out all negative values, which contradicts the intention of fully utilizing the expanded dimension. Regarding fixed regularization, we aimed to demonstrate that the method performs consistently well across datasets using the same regularization value. Otherwise, it would raise the concern of what would happen if we wanted to learn two datasets that require completely different regularization values at the same time.
>
> To further alleviate your concern, we re-ran both ACP and RanPAC with DINOv2 as the backbone, using $D=10000$ and ReLU for both methods. For ACP, we retained our fixed regularization. For RanPAC, we report results using both fixed regularization and its original cross-validation approach:
>
> | **Method**           | **CIFAR100 A_avg** | **CIFAR100 A_last** | **ImageNet-R A_avg** | **ImageNet-R A_last** | **CUB A_avg** | **CUB A_last** | **Cars A_avg** | **Cars A_last** | **T-ImageNet A_avg** | **T-ImageNet A_last** |
> |----------------------|:------------------:|:-------------------:|:--------------------:|:---------------------:|:-------------:|:--------------:|:--------------:|:---------------:|:---------------------:|:----------------------:|
> | ACP                  | 95.61              | 92.87               | 90.48                | 86.84                 | 93.48         | 90.66          | 93.52          | 90.51           | 92.13                 | 87.97                  |
> | RanPAC (fixed reg)   | 94.32              | 90.95               | 87.61                | 84.74                 | 91.85         | 88.94          | 91.81          | 87.31           | 89.66                 | 85.10                  |
> | RanPAC (cross-val)   | 94.47              | 91.13               | 88.52                | 84.74                 | 92.10         | 89.27          | 91.99          | 88.20           | 89.38                 | 84.25                  |
>
> The results show that cross-validation yields slightly better results overall, although it hurts performance on dataset T-ImageNet. We also note that ACP could be extended to use cross-validation if desired. More importantly, ACP continues to outperform RanPAC consistently across all datasets.
>
> - **Motivation to regress a class prototype:**
>
> If we understand your question correctly, you are asking why we regress the target prototypes instead of directly returning the class means as the output of the contrastive projection (CP) layer. If that is indeed the question, this approach would not work, since, contrary to the suggestion, we do not have an explicit mapping from the samples to the class means during inference, as the labels are not available. Therefore, directly outputting the class mean is not feasible at test time. Also, the target prototypes we regress are not exactly the original class means; the class means are shifted to improve their separation through Lemma 4.1. We are effectively learning a mapping that, without requiring the label during inference, improves the features by pulling samples from the same class together and pushing them away from other classes. Then, the classifier is learned on top of the improved features, as depicted in Figure 1, which is better than learning it directly on top of the original PTM features.
>
> - **Whitening and de-whitening steps:**
>
> Our goal is to improve class separability, which we quantify by the sum of pairwise cosine similarities between class means. It is indeed possible to apply SVD directly to $C$ and modify its singular values to reduce this sum. However, there is a key limitation with cosine similarity that we aim to address: cosine similarity does not take into account the scale or variance of each feature dimension. For example, in a dimension with low variance, even a small separation is significant, whereas in high-variance dimensions, a much larger separation is needed to achieve the same effect.
>
> To resolve this, we apply whitening transformation, which is a common data processing method. Whitening sets the variance of all feature dimensions to 1, allowing for a more meaningful measure of separation across directions. This process is closely related to Mahalanobis distance, which has been shown to outperform cosine similarity in related contexts (we refer the reviewer to FeCAM). By whitening the features, we essentially assess separation using Mahalanobis distance, and after increasing the separation, we de-whiten the features to restore the original variance. We will clarify this point further in the revision.
>
> - **Lemma 4.1 explanation:**
>
> The intuition behind Eq. 13 is that the diagonal shift is intended to perturb the class means in directions that reduce their pairwise cosine similarity. Specifically, each $w_i$ in the subspace $SV_T$ is perturbed by $\alpha \delta_i e_i$, where $\{e_i\}$ vectors form an orthonormal basis. This means each $w_i$ is shifted in an orthogonal direction by a magnitude of $\alpha$. The sign of $\delta_i$ determines whether the shift is in the direction of $e_i$ or the opposite. This condition arises from the proof of Lemma 4.1, where we take the derivative of the function $f_i(\alpha)$, representing the sum of cosine similarities for class $i$. By choosing $\delta_i$ in this way, we ensure that the slope of $f_i(\alpha)$ at $\alpha = 0$ is negative, guaranteeing that a small positive $\alpha$ will reduce cosine similarity.
>
> Regarding the choice of $\alpha$, if it is too small, the shift becomes negligible; if it is too large, the reduction in cosine similarity may not hold. While it is again possible to optimize $\alpha$ via cross-validation on each dataset, our goal was to show that a single value works across all datasets. We empirically selected $\alpha = 1$ by validating on CIFAR100 training data, searching over the range $[0.1, 1, 2, 5, 10]$. This value was then used consistently across all datasets without tuning.
>
> We agree that a more detailed explanation of Lemma 4.1 and its implications could enhance clarity, and we will include an expanded discussion in the revision.
>
> - **Suggested papers:**
>
> Thank you for the helpful suggestions. We will ensure that the recommended papers are cited and discussed in the revised manuscript, as well as incorporated into the experiments.
>
>    InfLoRA and HidePrompt are different approaches from ours. LoRanPac, which is more related to our work, projects the PTM features into a very high-dimensional space (e.g., 100,000) and solves the over-parameterized least-squares problem by truncating the singular value decomposition of the projected features. Their improvement over RanPAC seems to stem from the use of very high-dimensional projections. We note that this technique may also be used in our system to solve the extreme learning machine objective using 100,000 projection dimensions instead of our default of 5,000. Our contribution is entirely different: we propose an analytical contrastive layer that pulls samples of a class toward their mean and pushes class means away from one another, which improves class separation and consequently classification accuracy.
>
> - **Question 1:**
>
> As for Eq. 11, we clarify that $\Sigma^{-1/2}$ represents the whitening transformation, i.e., the inverse square root of the covariance matrix. For this to exist, $\Sigma$ must be positive-definite, which requires at least $n \geq d$ linearly independent data points. If this condition is not met (e.g., due to a small data size), we add a small regularization term $\epsilon I$ to the covariance to ensure it becomes invertible. In our implementation, we used $10^{-4} I$.

---

> > ### Comment · Reviewer_iEp4 · 2025-08-06
> > **Thanks for the rebuttal**
> >
> > Dear authors, thanks for responding to my comments. The concerns have been adequately addressed.

---

### Official Review · Reviewer_Ah55 · 2025-07-08

**Clarity:** 3
**Significance:** 3
**Originality:** 3
**Rating:** 5
**Confidence:** 4

**Summary:**

This paper introduces a method for class-incremental learning (CIL), called Analytical Contrastive Projection (ACP), which bridges the gap between the efficiency and stability of analytical methods and the adaptability of feature learning in continual learning settings. ACP leverages a pre-trained model (PTM) as a frozen feature extractor and proposes an analytical, non-gradient-based mechanism for feature adaptation using a Contrastive Projection (CP) layer. The method analytically aligns samples from the same class while increasing inter-class separation via prototype regression and class mean repulsion and integrates an ELM classifier for final prediction. The authors demonstrate through comprehensive experiments on several benchmark datasets that ACP not only outperforms prior analytical CIL methods and rehearsal-free baselines but also reaches or surpasses the performance of joint training in some cases. Ablation studies and efficiency analyses further support the method’s design and scalability.

**Questions:**

- Could you provide a more intuitive or geometric explanation of Lemma 4.1? A visual illustration showing how the angular separation between class means is improved would significantly enhance accessibility.
- Could you elaborate on why using a single shared covariance matrix across all classes works well for pseudo-replay? Have you observed any failure cases or dataset-specific limitations when using this approximation?
- Since the shared covariance is also used during pseudo-sample generation and classification, how are labels assigned to the generated samples, and how robust is this process across tasks?
- Would you consider updating or expanding Figure 1 to more clearly show how the matrices described in Section 4 are constructed and applied across tasks? For example, depicting the flow of Gram matrices, projection heads, and ELM updates could greatly help clarify the method.

**Ethical Concerns:**

["NO or VERY MINOR ethics concerns only"]

**Limitations:**

Yes

**Quality:**

3

**Strengths And Weaknesses:**

## Strengths

The paper addresses a highly relevant problem in CIL, while maintaining their computational advantages. The proposed ACP method is conceptually elegant and grounded in strong mathematical formulations, using closed-form solutions to approximate contrastive learning effects. The methodology is clearly described and logically structured. The experimental section is thorough, covering multiple datasets, PTMs (DINO-v2 and MoCo-v3), and a wide array of baselines including recent prompt-based and analytical methods. Results consistently show significant improvements in both average and final accuracy, with ACP performing on par with joint fine-tuning in many scenarios. The ablation studies are comprehensive and confirm the contributions of individual components like the CP layer, class mean repulsion, and the ELM classifier.

## Weaknesses

Despite its merits, a few limitations of the paper deserve further clarification. First, the analytical derivation of class mean repulsion using Lemma 4.1 is mathematically dense and lacks an intuitive geometric explanation or visual illustration, which may limit accessibility to a broader audience. Second, while the proposed pseudo-replay strategy is efficient and empirically effective, the justification for using a single shared covariance matrix across all classes is not fully addressed, especially considering that this shared covariance is used to generate pseudo samples. It remains unclear how the method distinguishes between samples from different tasks (e.g., task A vs. task B) during pseudo-replay, or whether such differentiation is even required under the proposed framework. A deeper discussion of why this approximation works and what its potential limitations are would be beneficial. Finally, although the methodology is described in detail, some parts of the pipeline remain difficult to follow without visual aid. In particular, the main figure (Figure 1) could be improved by clearly showing how the various matrices introduced in Section 4 (e.g., Gram matrices, projection matrices, and components of the pseudo-replay mechanism) are constructed and used during learning.

---

> ### Author Rebuttal · Authors · 2025-07-29
>
> - **Question 1:**
>   Intuitively, if class means are too close (i.e., having high cosine similarity), they become easily confusable, making the classification problem harder. Lemma 4.1 enables a transformation where class means are repelled from one another. We will consider including a more detailed geometric visualization in the revision. For now, we refer the reviewer to the t-SNE visualization in Figure 1 (right). The black dots represent class means. As shown, the class means of the "extracted features" are less separated, whereas after applying ACP, the data distributions and the class means in the "enhanced features" are more distinct, which helps classification.
>
> - **Question 2:**
>   The assumption behind our pseudo-replay mechanism is that each class distribution is characterized by its mean and a covariance matrix, an approach also used in prior work such as SLCA. However, maintaining a separate covariance matrix for each class increases the number of parameters. We instead use a shared covariance matrix across all classes, which drastically reduces the parameter cost while preserving the performance. This essentially implies that feature correlations are consistent across classes, i.e., if two feature dimensions are positively correlated for one class, they are likely to be so for others as well. In other words, the correlation belongs more to the feature extractor than to specific classes.
>
>   We found that this works well across all datasets we tested (see Tables 1 and 2), and did not observe any failure cases. This shared covariance assumption is also used in the classic Mahalanobis distance and linear discriminant analysis (LDA) methods. It has been shown to perform well in various settings.
>
> - **Question 3:**
>   Our approach does not rely on any task information for classification or pseudo-replay data generation. In CIL, tasks are often used to organize the sequence of class arrivals. We follow this in our experiments, but our method treats each class independently; we can learn a single class at a time. To generate a pseudo-sample for a class, we sample a noise vector from the shared covariance and add it to the corresponding class mean, which is then labeled with that class.
>
> - **Question 4:**
>   We appreciate the suggestion to enhance Figure 1. In the revised version, we will update the figure to better illustrate the construction and application of each component.

---

> > ### Comment · Reviewer_Ah55 · 2025-08-06
> >
> > I'd like to thank the authors for thoughtfully addressing my comments. The clarifications resolve my concerns, so I'm happy to keep my current score.

---

### Decision · Program_Chairs · 2025-09-17

**Decision:**

Accept (spotlight)

**Comment:**

This paper proposes Analytical Contrastive Projection (ACP), a novel class-incremental continual learning method that combines analytical closed-form updates with contrastive feature adaptation. The approach leverages frozen pre-trained models, a contrastive projection layer, and pseudo-replay with shared covariance, achieving strong results across multiple benchmarks. Reviewers praised the method’s conceptual elegance, efficiency, and thorough experimental validation, with ablations confirming the contributions of key components. Main concerns included the readability of Lemma 4.1, reliance on strong PTMs, fairness of some comparisons, and clarity of presentation; however, the rebuttal provided convincing clarifications, additional experiments, and commitments to improve the exposition. Overall, the reviewers agreed that the strengths outweigh the weaknesses, and the paper makes a solid and impactful contribution to continual learning. The simplicity and elegance of the method make it stand out of the other submissions, especially given how many papers achieve performance improvements at the cost of overcomplicated setups.